# Atomic-Level Sn Doping Effect in Ga_2_O_3_ Films Using Plasma-Enhanced Atomic Layer Deposition

**DOI:** 10.3390/nano12234256

**Published:** 2022-11-30

**Authors:** Yi Shen, Hong-Ping Ma, Lin Gu, Jie Zhang, Wei Huang, Jing-Tao Zhu, Qing-Chun Zhang

**Affiliations:** 1Institute of Wide Bandgap Semiconductors and Future Lighting, Academy for Engineering & Technology, Fudan University, Shanghai 200433, China; 2Shanghai Research Center for Silicon Carbide Power Devices Engineering & Technology, Fudan University, Shanghai 200433, China; 3Institute of Wide Bandgap Semiconductor Materials and Devices, Research Institute of Fudan University in Ningbo, Ningbo 315327, China; 4Institute of Precision Optical Engineering, School of Physics Science and Engineering, Tongji University, Shanghai 200092, China

**Keywords:** Ga_2_O_3_ film, Sn doping, plasma-enhanced atomic layer deposition, X-ray photoelectron spectroscopy, energy band alignment, electrical properties

## Abstract

In this work, the atomic level doping of Sn into Ga_2_O_3_ films was successfully deposited by using a plasma-enhanced atomic layer deposition method. Here, we systematically studied the changes in the chemical state, microstructure evolution, optical properties, energy band alignment, and electrical properties for various configurations of the Sn-doped Ga_2_O_3_ films. The results indicated that all the films have high transparency with an average transmittance of above 90% over ultraviolet and visible light wavelengths. X-ray reflectivity and spectroscopic ellipsometry measurement indicated that the Sn doping level affects the density, refractive index, and extinction coefficient. In particular, the chemical microstructure and energy band structure for the Sn-doped Ga_2_O_3_ films were analyzed and discussed in detail. With an increase in the Sn content, the ratio of Sn–O bonding increases, but by contrast, the proportion of the oxygen vacancies decreases. The reduction in the oxygen vacancy content leads to an increase in the valence band maximum, but the energy bandgap decreases from 4.73 to 4.31 eV. Moreover, with the increase in Sn content, the breakdown mode transformed the hard breakdown into the soft breakdown. The *C*-*V* characteristics proved that the Sn-doped Ga_2_O_3_ films have large permittivity. These studies offer a foundation and a systematical analysis for assisting the design and application of Ga_2_O_3_ film-based transparent devices.

## 1. Introduction

Gallium oxide (Ga_2_O_3_) has received increasing interest during the past decade owing to its intriguing physical properties, such as its excellent transparency [1], wide bandgap [2], high radiation resistance [3], high saturation electron velocity [4], and the ability for its large-scale single crystal manufacturing by the industrial-grade Czochralski method [5] and the Floating Zone technique [6]. Therefore, it can possess many applications, including photocatalysts, transparent conducting oxides [7], solar-blind UV detectors [8], high-power and high-voltage devices [9], and gas sensors [10]. It is reported that Ga_2_O_3_ materials contain five different polymorphs, which are the rhombohedral (*α*), the monoclinic (*β*), the defective spinel (*γ*), the cubic (*δ*), and the orthorhombic (*ε*) structures [11]. Among these five polymorphs, *β*-Ga_2_O_3_ is the most thermodynamically stable phase [12].

Many studies investigate the doping effects of different elements on their properties via experiments and first-principles calculations [13,14]. The group-IV; elements (Si [15], Ge [16], and Sn [17]) are essential doping elements for n-type *β*-Ga_2_O_3_. For the example of the Ge element, Ahmadi et al. grew Ge-doped Ga_2_O_3_ thin films using plasma-assisted molecular beam epitaxy [16]. It is found that the electron concentration and mobility are both improved by a Van der Pauw Hall analysis. Previous researches involve the preparation of Sn-doped thin films by plasma-assisted molecular beam epitaxy (MBE) [17,18], chemical vapor deposition (CVD) [8,19], metal-organic vapor phase epitaxy (MOVPE) [20], hydrostatic press [21], and so on; in these studies, the electrical performance is the primary concern. However, there is no systematic analysis of the change in chemical state, microstructure evolution, and the bandgap structure of the Sn-doped film. Additionally, the mechanism describing the influence of doping on the film properties remains unclear. Furthermore, although there have been many studies on the growth of undoped Ga_2_O_3_ film using atomic layer deposition (ALD) [22,23], few researchers have reported on the precise doping of Ga_2_O_3_ film via ALD [15,24].

ALD has self-limiting growth characteristics that can be employed in single atomic-scale deposition technology [25]. It can accurately tune growth during the preparation of functional semiconductors. By adjusting the thickness of the SnO_2_ sublayer, it can precisely control the concentration of the Sn atom which enables either doping with a minimal amount or a large amount of Sn atom in the Ga_2_O_3_ film. Currently, there have been few reports on Sn-doped Ga_2_O_3_ thin films prepared by the plasma-enhanced ALD (PEALD) method. In this paper, the PEALD technology was used to prepare a high-purity Sn-doped Ga_2_O_3_ film on quartz and Si substrates. The results from X-ray photoelectron spectroscopy (XPS) show the effects of Sn doping on the chemical bonding characteristics and the chemical composition of the obtained films. The different subpeak evolution of the O1s peaks is analyzed in detail for various Sn concentrations. Furthermore, we systematically studied the basic film characteristics with an atomic force microscope (AFM) and X-ray reflection (XRR). Then, the influence of the Sn doping was identified using spectroscopic ellipsometry (SE) and UV–Vis measurements on the optical constant and the light transmittance. The optical energy bandgap, *E*_g_, and the energy band alignment were defined. Finally, the electrical properties of the metal-oxide-semiconductor (MOS) structures with Sn-doped Ga_2_O_3_ films were examined to evaluate the effect of Sn doping on the breakdown electrical voltage and saturation capacitance (*C*_s_).

## 2. Experimental

### 2.1. Sample Preparation

Deposited Sn-doped Ga_2_O_3_ films with differing Sn concentrations were sequentially prepared with the BENEQ TFS200 ALD system (BENEQ Inc., Espoo, Finland) at 200 °C on two different substrates. Optical transmittance was measured in the samples grown onto a quartz substrate. The Si wafers were cleaned before the deposition of the Ga_2_O_3_ films; this process was described in previous work [15]. During the deposition process, trimethylgallium (TMG, 99.9999%, Fornano Inc., Suzhou, China) and tetrakis(dimethylamino)tin (TDMASn, 99.9999%, Fornano Inc., Suzhou, China) were used as the sources of gallium and tin, respectively. TMG and TDMASn were stored at 10 ℃ and 40 ℃ in a stainless bottle, respectively. The precursor for the oxygen was O_2_ plasma, which was activated at a power of 200 W.

The schematic diagram for the ALD super cycle of the Sn:Ga_2_O_3_ growth utilized in this work is shown in Figure 1, in which the individual TMG, TDMASn, and O_2_ are indicated. Every ALD cycle includes four processes: the TMG or TDMASn was pulsated for 50 ms or 2 s, the Ar was purged for 2 s, the O_2_ plasma processing was applied for 8 s, and the Ar was purged for 2 s. For one super ALD cycle, which includes N_1_ ALD cycles of Ga_2_O_3_ sublayer growth and N_2_ ALD cycles of SnO_2_ sublayer growth, the deposited film is composed of N super ALD cycles. The Sn-doped Ga_2_O_3_ films with different Sn contents were prepared by control of the ratio of N_1_ and N_2_ during one super cycle. The thickness of the films is an important parameter that influences the properties of the sample. Therefore, it is necessary for all the films to be kept at the same thickness. In short, the total number of ALD cycles has remained consistent throughout our experiments. For Ga_2_O_3_ thin films with various contents of Sn, the numbers of N_1_, N_2_, and N for the deposited parameters are summarized, which are listed in Table 1.

### 2.2. Characterization of the Sample

The chemical composition of the obtained films was determined by the use of X-ray photoelectron spectroscopy (XPS, Thermo Scientific^TM^ K-AlphaT^M+^, Thermo Fisher Scientific Inc., Waltham, MA, USA) that involved a monochromatic Al Kα source (1486.6 eV). To remove the small amount of adventitious carbon and ambient water vapor on the surface, the films were etched by Ar ion for 5 nm thickness. The step sizes in the XPS were 1 eV, 0.1 eV, and 0.02 eV for the survey spectrum, the core level spectra, and the valence band maximum (VBM), respectively. All the peaks would be referenced with respect to the C1s peak binding energy at 284.8 eV for adventitious carbon [26]. The surface topography and the microstructure of the films were examined by atomic force microscope (AFM, Bruker icon, Bruker Inc., Billerica, MA, USA) and by X-ray reflection (XRR, Bruker D8, Bruker Inc., Billerica, MA, USA). The SE spectral measurements were performed at 75° using a rotating analyzer ellipsometer (SOPRA GES-5E, SOPRA Inc., Annecy, France). This setup was used to measure the spectra of *Ψ* and *Δ* as functions of the wavelength, *λ*. Fittings on the resulting spectra were created with WinElli_II software (SOPRA Inc., Annecy, France). Using ultraviolet-visible spectroscopy (UV–Vis, UV–3100, Shimadzu Inc., Kyoto, Japan) over the wavelength range of 200–1000 nm, transmittance–wavelength curves were obtained for all of the films deposited onto the quartz substrates.

## 3. Results and Discussion

### 3.1. Chemical Composition

Figure 2 shows a survey of the XPS spectrum for the as-deposited PEALD Sn-doped Ga_2_O_3_ thin films with Sn contents of 0%, 1%, 5%, 10%, 20%, and 50%. The XPS wide scan spectra were dominated by the signals for the gallium (Ga 2s, Ga 2p, Ga 3s, Ga 3p, Ga 3d), the stannum (Sn 3p, Sn 3d), and the oxygen (O 1s) alongside the Auger peaks from the gallium (Ga LM) and the oxygen (O KL1) in Figure 2a; all of which are in agreement with those reported in the literature [15,27]. These results represent an excellent purity for the Sn-doped Ga_2_O_3_ thin films. Due to their close binding energy, the XPS peak for the Sn 3d is difficult to discern from the Auger peaks of the Ga LM [28]. Notably, the Sn 3p peaks slowly evolve with increased Sn composition, but they are too weak to be detected. The detailed XPS scans of the Sn 3p_3/2_ component are shown in Figure 2b. As the composition of Sn increases, the intensity of the Sn 3p_3/2_ peak is significantly improved, suggesting that the Sn is successfully doped into the Ga_2_O_3_.

Moreover, the Ga 2p and O 1s peaks were scanned with a high resolution for different Sn contents, respectively, as shown in Figure 2c,d. The increase in the content of Sn causes a tiny offset in the binding energy position of the Ga 2p in Figure 2c. This is because the Sn atoms replace some sites of the Ga atom, which leads to this decrease in the binding energy of the Ga 2p, whereas the energy difference between the Ga 2p_3/2_ and Ga 2p_1/2_ components is unchanged with an increase in the Sn composition; this difference still maintains about 26.9 ± 0.1 eV. As the proportion of Sn^4+^ replacing Ga^3+^ increases, the peak position of the O 1s shifts towards lower binding energy. The latter is discussed in more detail later.

To trace the influence of the doping concentration of Sn on the chemical valence and the composition, a deconvolution procedure for the O1s XPS peaks is performed [29], as shown in Figure 3. In the O 1s peak of the XPS spectrum, the signal at 532.7 eV is not observed [30], which means that adsorbed molecular water does not exist. In the Ga_2_O_3_ thin films with no Sn doping, two different subpeaks are observed at 532.1 eV and 531.3 eV. The major constituent is located at the binding energy around 531.3 eV, which relates to the Ga–O bond [31], whose area occupies about 75.8% of the total area of the O 1s peak. The other portion is generally allocated to the oxygen vacancies in the higher binding energy around 532.1 eV, defined as O_v_ [32,33]. For the Sn-doping Ga_2_O_3_ thin films, the O 1s peak is split into three or four subpeaks. The peaks at 530.5 eV and 529.3 eV are assigned to the Sn–O bond in various chemical states, i.e., SnO_2_ and SnO, respectively [34]. The Sn–O bonds of SnO_2_ and SnO components are defined as Sn–O_I_ and Sn–O_II_, respectively. In all the Sn–O binary systems, only SnO_2_ and SnO are known to be thermodynamically stable [35]. As the content of Sn increases, the XPS intensity of the subpeak relating to Sn–O_I_ and Sn–O_II_ increases. Meanwhile, there is a decrease in the proportion of the Ga–O bonds. These bonds are replaced by Sn–O_I_ or Sn–O_II_ bonds, suggesting that the Sn atoms occupy some of the Ga sites in the Ga_2_O_3_ compound [28]. That means the stoichiometry of the Ga_2_O_3_ film can be tuned by modifying the composition of the Sn doping. It is noteworthy that the Sn–O_II_ appears at the 20% and 50% contents of Sn doping. This may be related to the lower Gibbs free energy for SnO_2_ formation compared to that of SnO, which indicates that it is easier to form SnO_2_ than SnO [36]. Only at high concentrations are small amounts of SnO formed. Figure 3g–i shows the percent of O_v_, Ga–O, and Sn–O bonds as a function of the Sn content, respectively. We observe here that all of these values are consistent with the monotonic linear trend with an increase in the Sn composition. As the Sn content increases from 0% to 50%, the Sn atoms begin to occupy the Ga sites and the oxygen vacancies, which leads to a reduction in the proportion of Ga–O and O_v_. Noticeably, the proportion of Sn–O is identical to that of Sn doping, proving that the concentration of Sn doping is effectively regulated by the super cycling method.

### 3.2. Morphology and Microstructure

The AFM images of Figure 4 show that the surface features and the morphology of the Ga_2_O_3_ films vary with a change in the content of Sn. All of the as-deposited Ga_2_O_3_ films display an acicular structure, which is similar to other reported amorphous materials [37,38]. Moreover, it is determined that the surfaces of the thin films deposited via PEALD are smooth, which suggests that a deposition is extremely uniform. The related root-mean-square (RMS) roughness values are observed to be from 0.2 nm to 0.9 nm for pure Ga_2_O_3_ film, and the RMS roughness of the films with different Sn concentrations is counted in Figure 5b. Noticeably, with an increase in Sn content, the grain size and the surface roughness become larger. This may relate to the fact that the RMS roughness of the SnO_2_ films is larger than that of the Ga_2_O_3_ films [39], so the variation in the RMS roughness corresponds to an increase in the Sn composition [8].

Figure 5 presents the XRR results for the Ga_2_O_3_ films with different Sn concentrations. The simulated XRR spectrum (curves) was obtained via a bilayer model composed of a Ga_2_O_3_ layer and a thin SiO_2_ buffer layer. The simulated curves are consistent with the data for all the films with 2*θ* in the range of 0.6° to 4.0°. In addition, it is evidenced that the films have excellent uniformity and smooth interfaces with well-developed oscillations [40]. Furthermore, the periods of the Kiessig fringes [41] of the sample are around 0.3° for the Ga_2_O_3_ films with 50% Sn compositions, about 0.4° for Sn composition at 0 and 5, and roughly 0.5° for Sn compositions at 1, 10, and 20%. This means that the film thickness with 0% and 50% Sn contents are larger than that of all the other samples. Moreover, the film thicknesses obtained from the XRR and SE data, as a function of the Sn composition, are plotted in Figure 5c; these evince a similar variation tendency in thickness. When the Sn composition is less than 20%, the RMS roughness obtained by the AFM is almost lower than the value simulated from the XRR data, as shown in Figure 5b. If the Sn composition is further increased to 20 and 50%, the RMS roughness of the AFM is larger than that of the XRR date. The reason for this may be the ability of the XRR to obtain in-depth information inside the film, which includes a large number of details on the roughness of the SnO_2_ sublayers [39]. As shown in Figure 5d, the densities were measured from the XRR for the Ga_2_O_3_ films with different Sn contents. Here, the density of the non-doped Ga_2_O_3_ film is 5.7 g/cm^3^, which is lower than the Ga_2_O_3_ bulk material [42]. This discrepancy could be ascribed to the fact that the Ga_2_O_3_ film, processed via PEALD, is amorphous at low deposition temperatures, and it is not compact. Since the mass density of the SnO_2_ bulk material is around 6.7–6.9 g/cm^3^ [43,44], the density of the Sn-doped Ga_2_O_3_ film increases as the content of Sn increases to 5%. The other reason could be that some Sn ions may act as interstitial atoms located in the lattice at low doping levels [19]. It is noteworthy that the density dramatically decreases with a composition of 10% Sn. Such a trend may be caused by the appearance of tiny and non-dense voids at the interface, which is formed from the roughness of the SnO_2_ sublayers. With the further rising of Sn content, the proportion of SnO_2_ film with a larger density is stepped up, which makes the density of the Sn-doped Ga_2_O_3_ film increase.

### 3.3. Optical Properties

By use of the SE fitting data, the refractive indices *n* of the Sn-doped Ga_2_O_3_ films can be derived, and they are plotted in Figure 6a. The non-doped Ga_2_O_3_ film reveals a high *n* value of 1.93 at a wavelength of 700 nm, which is consistent with previous reports [45,46]. When the level of doping increases, except for 50% Sn, there is a monotonic decrease in the refractive index. This may occur since an Sn impurity can play the part of effective donor-generating free carriers [47,48]. Sn doping increases the concentration of free carriers in the films, which results in a diminution in their refractive index [49]. At a 50% content of Sn, the refractive of the film is the highest compared with the others, which is an abnormal phenomenon. This is mainly attributed to the high *n* value for the pure SnO_2_ of approximately 1.99 [39]. At a high level of doping, the ratio of SnO_2_ in the multilayer plays a major role in the creation of the increased refractive index of the Sn-doped Ga_2_O_3_ film. In addition, when too many Sn^4+^ ions are doped into the Ga_2_O_3_ film, the high Sn concentration deteriorates the quality of the film and increases the number of defects inside, which leads to an increased *n* value [50]. In addition to the effect of the change in the Sn content on the inherent optical properties of the films, the roughness of the film surface is also an important indicator of the optical properties of the film [51,52]. An increase in surface roughness leads to a decrease in film reflectivity [51]. This is also in approximate agreement with our experimental phenomena. Figure 6b presents the extinction coefficient *k* of the films with various doping levels. A rough film surface causes light to be diffusely reflected onto the surface of the film, so the increase in surface roughness causes the extinction coefficient of the film to become larger [53]. Between the wavelengths of 220 nm–260 nm, the extinction coefficient of 50% Sn content is the largest, while 0% is the smallest, which may also be related to the roughness of the film surface. Over the entire wavelength range of 340–800 nm, the values of *k* for all the films are close to zero; thus, it is demonstrated that the films are almost transparent in this wavelength region. This feature could provide important applications in the field of transparent semiconductor oxides (TSO) [10,32,54]. The inset in Figure 6b shows a schematic diagram of the layer structure, which is used as the basis for the establishment of the elliptic fitting model.

The transparency of a film is an important feature that enables the fabrication of fully transparent electronics [2,32]. With this in mind, Figure 6c presents the optical transmission spectra versus the wavelength for the as-deposited Ga_2_O_3_ film at various contents of Sn. All of these films are highly transparent in the visible range, which highlights that the Ga_2_O_3_ films (processed via PEALD) could potentially be implemented in applications requiring transparency [55]. It is especially noteworthy that the pure Ga_2_O_3_ film reaches almost 100% transparency. As the Sn content increases, the transmittance decreases; even so, this value never dips below 90%. The average transmittance of the films, for the wavelength range 300–1000 nm, is shown in the inset of Figure 6c. It monotonically reduces from 98.0% to 93.6% for Sn-doped films when the Sn composition improves from 0% to 50%. The increase in transmission is directly related to the decrease in the extinction coefficient of the films. The transmittance of the film is directly related to the reflectance and extinction coefficient. In Figure 6b, the extinction coefficient decreases rapidly to close to 0 at a wavelength of 340 nm. Beyond 340 nm, the transmittance is mainly related to the reflectance of the film. At approximately 250 nm, the absorption edge of all the curves is visible. The intrinsic absorption process occurs by the electron transition from the O 2p states in the valence band maximum (VBM) and the Ga 4s states in the conduction band minimum (CBM) [56]. The effects of the various contents of Sn on VBM and CBM are discussed in Section 3.4. Additionally, we find that the caused defective levels could influence the range of absorption, such as the oxygen vacancies. The energy bandgap, *E*_g_, is an important parameter used to characterize semiconducting materials since it directly influences the electronic band structure of a material and the corresponding properties of a device [46,57]. Thus, studying the effects of the Sn composition on the *E*_g_ is necessary for the Ga_2_O_3_ film. Through the interpolation of the linear fit towards the absorption edge, the optical bandgap value can be obtained [47]. The relationship between the absorption coefficient and *E*_g_ is given by (*αhv*)^2^ ∝ *hv–E*_g_, where *α* is the optical absorption coefficient and *hv* is the photon energy [58]. Thus, a plot of (*αhv*)^2^ against *hv* can be used to determine the value of *E*_g_, as shown in Figure 6d. The bandgap of the pure Ga_2_O_3_ film is defined as ~ 4.84 eV, which is consistent with the previously reported value of these materials in the literature [59]. The values of *E*_g_ for the Sn-doped Ga_2_O_3_ films are 4.8 eV, 4.74 eV, 4.68 eV, 4.63 eV, and 4.52 eV for the Ga_2_O_3_ film with Sn contents of 1%, 5%, 10%, 20%, and 50%, respectively. Therefore, the *E*_g_ decreases as the Sn^4+^ cations replace the Ga^3+^ cations. In contrast, the *E*_g_ value is 3.59 eV in pure SnO_2_ materials, which is lower than that of the Ga_2_O_3_ materials [39]. In summary, with a change in the amount of Sn dopant used, the effective control of the energy bandgap in Ga_2_O_3_ films is demonstrated.

### 3.4. Energy Band Structure

Moreover, to determine the *E*_g_ of these Sn-doped Ga_2_O_3_ films, the O 1s energy loss spectrum is obtained by XPS. The energy loss structure on the high binding energy side of the O 1s core-level XPS peak was utilized to obtain the band gap energy through a linear fitting [60,61], as shown in Figure 7a. The *E*_g_ values are ~4.73 eV, ~4.69, ~4.66 eV, ~4.60 eV, ~4.44 eV, and ~4.31 eV for 0%, 1%, 5%, 10%, 20%, and 50% contents of Sn in the doped film, respectively. Usually, the direct bandgap progression for the alloyed compound A_1−*x*_B*_x_* is a good estimate determined from the relationship *E*_g_ = (1 − *x*)*E*_g_^A^ + *xE*_g_^B^−*bx*(1 − *x*) [62,63], in which *b* is called the bandgap bowing, and it is always a positive constant. For this work, we use the *E*_g_^A^ value of 4.73 eV for the pure Ga_2_O_3_, which is obtained by XPS analysis, and the *E*_g_^B^ is 3.59 eV for the pure SnO_2_, taken from the literature [39]; *x* is estimated by use of the XPS analysis in Figure 3i. In most cases, the value of *b* is usually both a teeny value and composition-independent [63,64]. The origin of the bandgap bowing is usually more complicated. Here, we approximate the effects of Sn doping on the trends in the bandgap variations, so the value of *b* is presumed to be zero. The estimated *E*_g_ is plotted in Figure 7c, which also shows a determination of the *E*_g_ values of the Ga_2_O_3_ film using UV–Vis and XPS data. Comparing the *E*_g_ values acquired by these three methods, it is determined that such values show a similar trend: the bandgap energy of the Ga_2_O_3_ films decreases with an increase in the Sn composition. Comparing the Si and Ge doping, Sn-doped Ga_2_O_3_ can reduce the energy band gap to cope with different application scenarios.

Figure 7b displays the variation of the valence band spectra between −4 and 15 eV for the Ga_2_O_3_ film as a function of the Sn ratio. The VBM was determined by the tangent method [24]. The minimum value of VBM, *E*_V_, is positioned at 2.64 eV for the pure Ga_2_O_3_ film. The VBM rises as the Sn concentration increases within the Ga_2_O_3_ films. The reason for this could be the reduction in the oxygen vacancy content (Figure 3g). When the Sn content reaches 50%, the VBM moves to 3.46 eV.

For the optoelectronic application of these Sn-doped Ga_2_O_3_ materials in LEDs, solar cells, or photodetectors, we are required to understand the nature of their VBM and CBM. The VBM is mainly composed of oxygen 2p-orbitals, whereas the metal-cation-vacant s-orbitals primarily constitute the CBM, which involves either Ga or Sn ions [65]. Optoelectronic applications are mainly governed by two important parameters that strongly depend on the nature of the VBM and CBM [66]: the charge transfer (injection or ejection) at the interface and the optical transitions (absorption or emission). Nevertheless, some aspects such as the energy of the VBM and CBM changing with varying Sn composition do not yet clearly show a diverging trend for the Ga_2_O_3_ thin films. The CBM is obtained using the expression *E*_C_ = *E*_V_ − *E*_g_, where *E*_g_ is obtained from the XPS data. Next, the energy band alignments of the Sn-doped Ga_2_O_3_ films with different Sn ratios are obtained, as shown in Figure 7d. The CBMs are −2.09 eV, −1.87 eV, and −0.85 eV for Ga_2_O_3_ with 0%, 10%, and 50% Sn contents, respectively. The surface Fermi level, *E*_f_, for the pure Ga_2_O_3_ film, is 2.09 eV below the CBM and 2.64 eV above the VBM. The Ga_2_O_3_ film with 1% Sn content exhibits similar results, in which the surface *E*_f_ is 2.03 eV below the CBM and 2.66 eV above the VBM. As the Sn content rises, both the changes of VBM and CBM have a similar tendency; however, the increase in VBM is less than that in CBM, which leads to the decrease in *E*_g_. With increased Sn content, the main contribution to the decline in the CBM is SnO_2_ with a low CBM value of 0.8 eV [67]. Moreover, Kavan reported that the *E*_C_ value of the ALD-made amorphous SnO_2_ is downshifted by 0.25 eV compared to other methods [68]. Such basic band information for the Sn-doped Ga_2_O_3_ film will assist in the design of suitable interfaces for both hetero-structured nanocrystals and potential devices.

### 3.5. Electrical Properties

To investigate the electric properties of the MOS structures with Ga_2_O_3_ films, leakage current-voltage (*I*-*V*) and capacitance-voltage (*C*-*V*) tests were performed on the as-prepared samples, as shown in Figure 8. The Cr/Au top electrodes, a thickness of 200 nm and 100 μm in width square shape arrays were deposited onto the samples. In Figure 8a, as the Sn content increases, Sn^4+^ cations replace some of the Ga^3+^ cations, and the curves shift left at the pre-breakdown region, displaying the critical breakdown electric voltage decreasing. At the 20% and 50% Sn contents, the samples exhibit visible soft breakdown, demonstrating that the breakdown mode transitions from hard (intrinsic) breakdown to soft breakdown. The reason for this may be tiny and non-dense voids inside the nanostructure [69], which is proved in Section 3.2. The other reason may be attributed to the rise in free charge carrier concentration as a result of extra electrons being contributed by the donor Sn^4+^ cations incorporated as replacement ions for Ga^3+^ cations [70]. Moreover, the leakage current density of all samples is less than 3 × 10^−6^ A/cm^2^ at 2 V, indicating that the good insulating property is suitable for dielectric and passivation materials for electronics.

The *C*-*V* curves of the MOS structures with PEALD Ga_2_O_3_ films were measured at the frequency of 1 MHz, as shown in Figure 8b. Due to the P-type substrate, the capacitance reaches a saturation maximum at the negative bias voltage. At voltages higher than −0.5 V, a modulation portion is observed in the *C*-*V* characteristics, and the capacitances decrease abruptly as the voltage increases. When the Sn content is less than 10%, the curves show the consecutive increase in saturation capacitance (*C*_s_) as the Sn content increases. Both slopes of the *C*-*V* characteristics in the modulation portion and *C*_s_ decrease for the Ga_2_O_3_ film with a 50% Sn content. However, the thickness of these films is different; the *C*_s_ cannot be directly compared, so the permittivity (*ε*) was calculated based on the thickness of these films measured by SE. The *ε* of Ga_2_O_3_ films with different Sn compositions is shown in the inset of Figure 8b. The *ε* values are ~9.88, ~10.14, ~12.12, ~14.01, ~14.07, and ~13.18 for 0%, 1%, 5%, 10%, 20%, and 50% contents of Sn in the doped film, respectively. When the Sn content is less than 10%, it shows a linear fit on the permittivity with Sn composition. Due to the improvement of the Sn-doped concentration, the *E*_g_ of the Ga_2_O_3_ film descends. This causes the reduction in an energetic barrier upon the application of an electric field. This makes it easier to allow electron injection from a metal into the conduction band of an insulator [71]. With the further increase in Sn content, although the *E*_g_ is further reduced, it leads to some defects at the interface, making the capacitance drop instead, especially at the 50% Sn content. For RF devices, it is important to improve the intrinsic capacitances to obtain better performance. Hence, it is an effective method for the precise regulation of thick SnO_2_ interlayers and nanostructures in Ga_2_O_3_ films to achieve controlling electrical properties.

## 4. Conclusions

In this work, various Sn-doped Ga_2_O_3_ films were prepared by the use of PEALD. It is proven that these films have a smooth surface and high quality, and the Sn atoms were successfully doped into them; these factors were determined by both XPS and AFM measurements. It is also found that the chemical state and the microstructure vary with a change in Sn doping content. The XPS analysis showed that the O 1s peaks of the Sn-doped Ga_2_O_3_ thin film consist of Ga–O, O_v_, Sn–O_I_, or Sn–O_II_ bonds. With an increase in the Sn dopant content, the Ga–O and O_v_ are gradually replaced by Sn–O_I_ and Sn–O_II_. Moreover, the density can reach the highest value of 6.23 g/cm^3^ at 5% Sn doping. The refractive index monotonically decreases from around 1.9 to 1.8, except for the 50% Sn doping, owing to the major contribution of SnO_2_, while the value of *n* is sharply boosted to 1.94. Furthermore, the average transmittance of all the films is over 90% in the wavelengths of 300–1000 nm and the energy bandgap of the Ga_2_O_3_ film decreases from ~4.73 eV to ~4.31 eV as the concentration of Sn in the film increases. Moreover, the VBM and CBM of the Sn-doped Ga_2_O_3_ film were analyzed by a valence band spectra method combined with the bandgap. The results indicate that the VBM and CBM increase as the Sn ratios grow. We also elucidate how the Sn composition contributes to the VBM and CBM of the Ga_2_O_3_ film. Considering the electrical aspect, the *I-V* and *C*-*V* tests were analyzed by employing MOS capacitors. The results show that the breakdown electrical field reduces with increasing Sn composition. The extra electrons from the donor Sn^4+^ cations and tiny voids cause soft breakdowns in the MOS structure. The permittivity of the Ga_2_O_3_ films can be tuned from 9.88 to 14.07 with the increase in Sn composition. However, 50% Sn doping reduces the permittivity due to some defects at the interface. Overall, our systematic and detailed research could help to fully utilize Sn-doped Ga_2_O_3_ films in various applications. These results may act as a guide towards improved high-performance devices, such as high-power and high-voltage devices, gas sensors, photoelectrical detectors, and various other possibilities.

## Figures and Tables

**Figure 1 nanomaterials-12-04256-f001:**
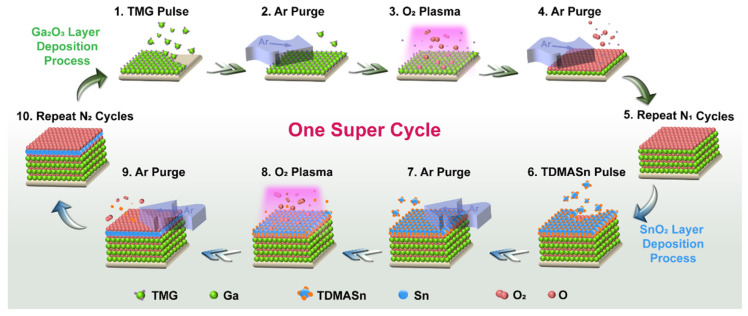
The schematic diagram for the ALD super cycle of the Sn-doped Ga_2_O_3_ film growth.

**Figure 2 nanomaterials-12-04256-f002:**
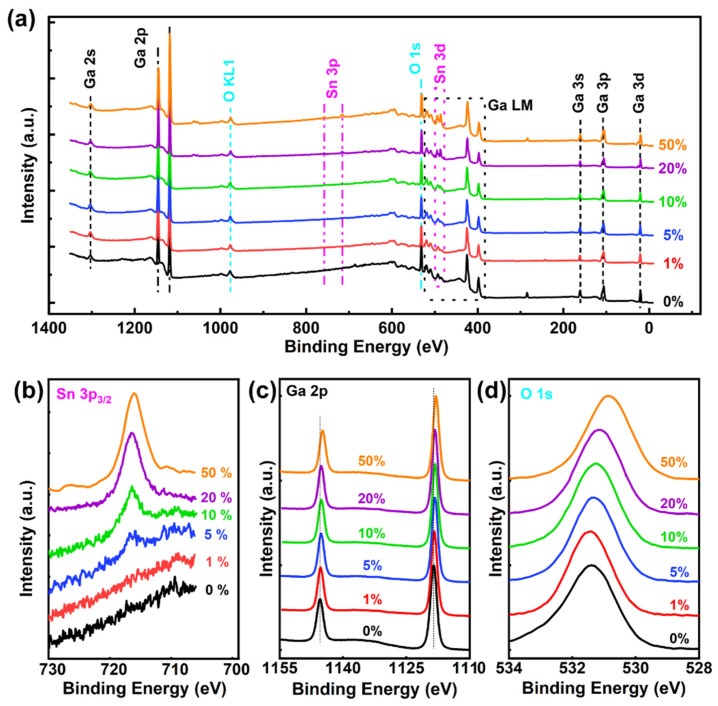
Graphs depicting the results of the XPS analysis for the Ga_2_O_3_ thin films with 0%, 1%, 5%, 10%, 20%, and 50% Sn compositions: (**a**) survey peaks, (**b**) Sn 3p_3/2_ spectra, (**c**) Ga 2p spectra, and (**d**) O 1s spectra.

**Figure 3 nanomaterials-12-04256-f003:**
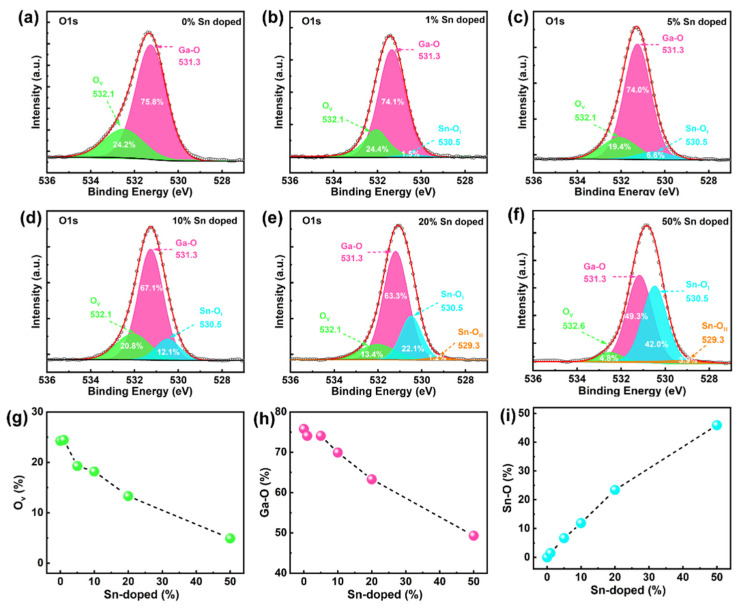
(**a**–**f**) XPS analysis of the O 1s peak of the Ga_2_O_3_ thin films for different concentrations of Sn. Graphs depicting the percent of (**g**) O_v_, (**h**) Ga–O, and (**i**) Sn–O bonds for various Sn concentrations.

**Figure 4 nanomaterials-12-04256-f004:**
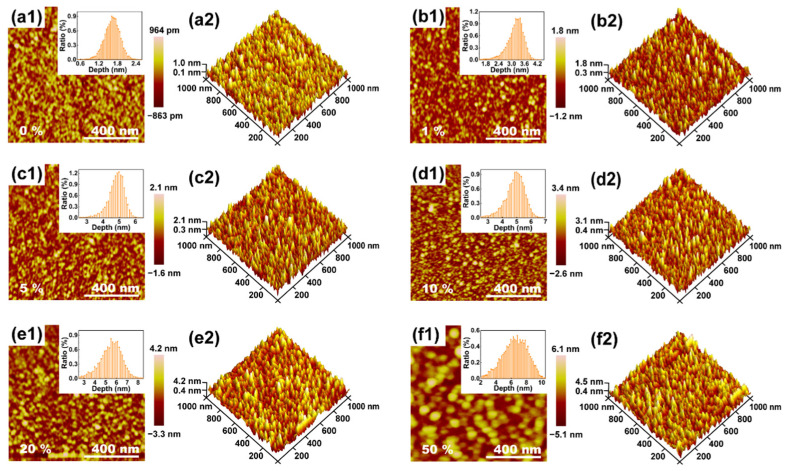
AFM representations of the 2D and 3D morphology of the Ga_2_O_3_ thin films with (**a1**,**a2**) 0%, (**b1**,**b2**) 1%, (**c1**,**c2**) 5%, (**d1**,**d2**) 10%, (**e1**,**e2**) 20%, and (**f1**,**f2**) 50% Sn content. The corresponding depth distribution images are shown in the inset of (**a1**–**f1**).

**Figure 5 nanomaterials-12-04256-f005:**
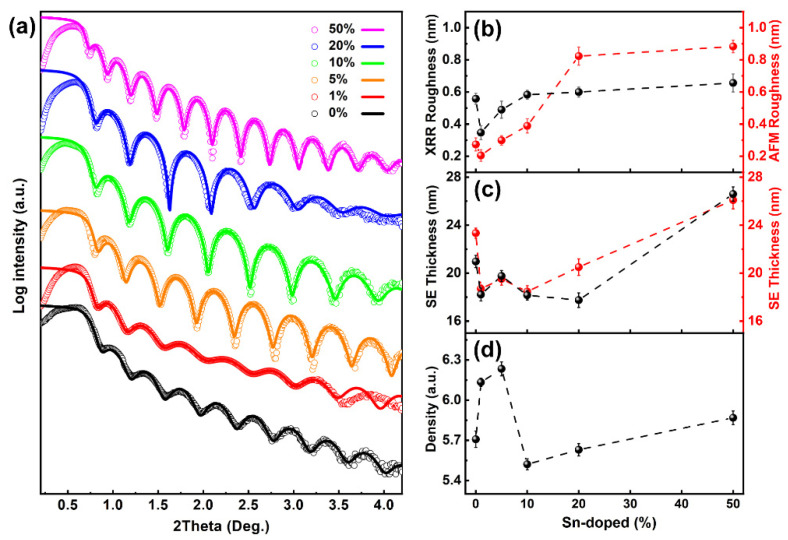
(**a**) Measured XRR (hollow cycle) for the Ga_2_O_3_ film with different Sn concentrations compared to the simulated data (curves). (**b**) The RMS roughness, (**c**) the thickness, and (**d**) the density of the film with variation in the Sn concentration.

**Figure 6 nanomaterials-12-04256-f006:**
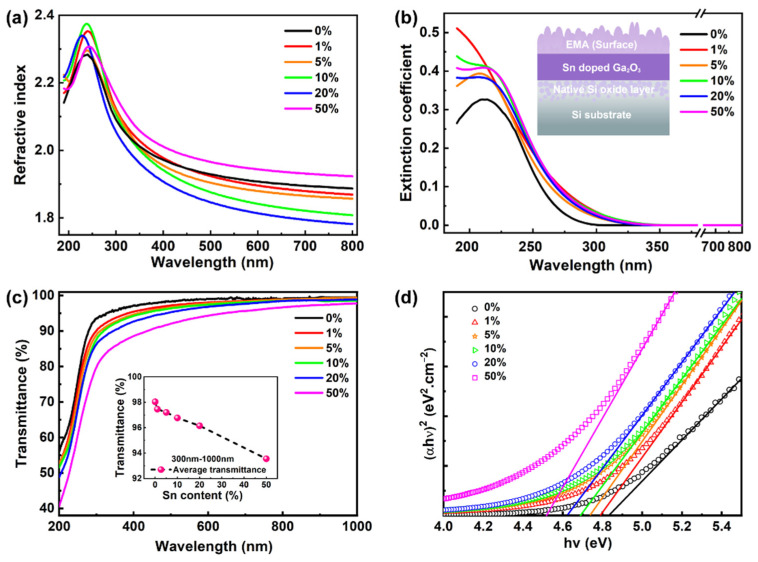
(**a**) Refractive index and (**b**) extinction coefficient for the various Sn-doped Ga_2_O_3_ thin films as a function of the wavelength. The inset in (**b**) depicts the schematic representation of the layer structure. (**c**) Transmittance spectra of the as-deposited Ga_2_O_3_ film at various contents of Sn. The inset displays the variation in the transmission with the content of Sn. (**d**) The (*αhν*)^2^ vs. *hν* curves are determined from the UV–Vis method. The extrapolation of the linear regions to the horizontal axis provides estimates of the *E*_g_ values.

**Figure 7 nanomaterials-12-04256-f007:**
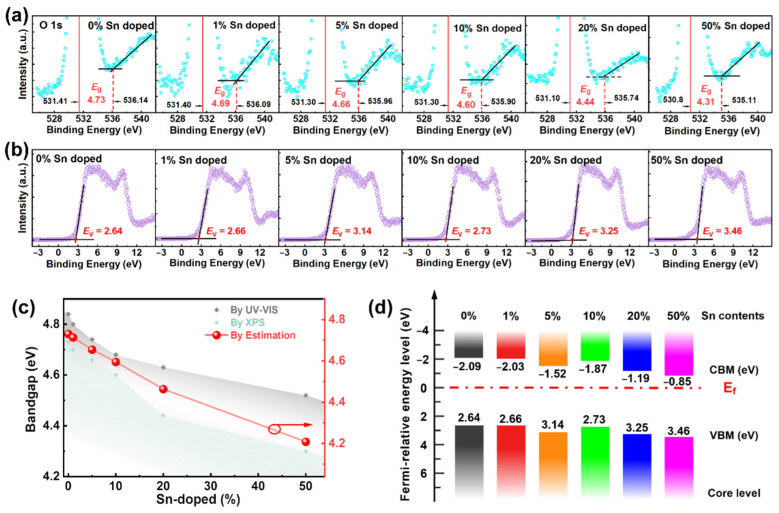
(**a**) Values of *E*_g_ found via the O 1s peak analysis using XPS measurements for Ga_2_O_3_ thin films doped with different Sn concentrations. (**b**) XPS valence-band spectra and VBM of the Ga_2_O_3_ thin films with varying content of Sn. (**c**) Change in the value of *E*_g_ as a function of the Sn content, determined through XPS measurements and the UV–Vis method. (**d**) VBM and CBM energy alignments for the Ga_2_O_3_ thin films as a function of Sn concentration.

**Figure 8 nanomaterials-12-04256-f008:**
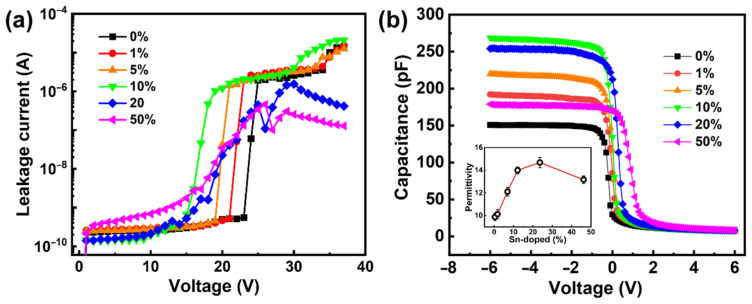
(**a**) *I*-*V* curves and (**b**) *C*-*V* curves of the Ga_2_O_3_ films with different Sn compositions. The inset of Figure 8b displays the variation in the permittivity with the content of Sn.

**Table 1 nanomaterials-12-04256-t001:** Deposition parameters for the Ga_2_O_3_ samples with various contents of Sn.

Sample	Sn Content	Ga_2_O_3_ Cycles (N_1_)	SnO_2_ Cycles (N_2_)	Super Cycles (N)	Total Cycles
1	0%	1	0	400	400
2	1%	99	1	4	400
3	5%	19	1	20	400
4	10%	9	1	40	400
5	20%	4	1	80	400
6	50%	1	1	200	400

## Data Availability

The data presented in this study are available on request from the corresponding author.

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
