# Peer review of "Atomic-Level Sn Doping Effect in Ga_2_O_3_ Films Using Plasma-Enhanced Atomic Layer Deposition"

_nanomaterials, 2022, doi:10.3390/nano12234256_

Round 1
Reviewer 1 Report
The authors grew Sn doped Ga2O3 films by plasma ALD. The Sn:Ga ratio was varied by varying the number of Sn and Ga layers while keeping the total layers constant. They then characterize the samples using various standard techniques. Lots of claims are not substantiated by their results.
1. XPS. Why did the authors analyze O 1s to determine Sn’s oxidation state? Why didn’t they examine the binding energy of Sn? Why does Sn 3d look the same regardless of Sn concentration?
2. AFM. The authors claimed the films were amorphous based on AFM images. AFM cannot tell you whether a film is amorphous or crystalline. 50%Sn sample show some nanoparticle structure. Is there compositional difference between the bumps vs. the background (flatter part)?
3. SE. How did the authors fit the SE results? Psi-delta? Did the authors fit for n, k, and thickness together? How good were the fits? It was claimed that “except for 50 % Sn, there is a monotonic decrease for the refractive index.” Another way to look at the data is that the lower index films have smaller thicknesses. (Fig. 6a and 5c) So the SE analyses might not be trustworthy.
4. UV-vis. The authors need to analyze the transmittance in terms of n and k. For wavelengths longer than 350 nm, these films do not absorb. Do the n values match the transmittance?
5. Density. The drop in density between 5 and 10% Sn is hard to understand. (Fig. 5d) The authors proposed voids. If that were the case, one would expect a corresponding increase in thickness and roughness, which are not seen in Fig. 5b and 5c.
6. Doping. What do authors mean by “atomic-level doping” in the title and abstract? Aren’t all doping atomic level by definition? How did doping occur in these samples? Were they annealed post deposition? Is the doping uniform? It was claimed that it’s n doping, but the paper did not provide any direct evidence. Neither J-E or C-V tells us the sign of doping. At the very least, the authors need to discuss what kind of doping one can expect from Sn4+ replacing Ga3+.
Other problems:
1. The paper is full of grammatical mistakes and typos.
2. The x-axis of Fig. 5b-d needs to be linear in Sn concentration, similar to Fig. 3 and 7.
Reviewer 2 Report
The study on "Atomic-Level Sn Doping Effect in Ga2O3 Films Using Plasma-2 Enhanced Atomic Layer deposition" is interesting and clearly presented.
Please consider some minor aspects:
Ceck the sentence: This may be related to the lower Gibbs free energy for SnO2 formation compared to that of SnO, which indicates that the formation of SnO is favorable at the higher concentration of Sn.
How can you explain the change in thickness since the total number of ALD cycles has been maintained?
Please include error bars or information on the thickness, density and permittivity estimations
Reviewer 3 Report
A report of reviewing a manuscript, nanomaterials-2044122
This manuscript reports (i) the film preparation, (ii) surface morphology, (iii) chemical bonding characteristics, (v) optical properties, and (iii) I-V and C-V characteristics of the atomic layer deposited films of Ga2O3 doped with various concentrations of Sn. The experimental results obtained were comprehensively and precisely interpreted to reveal the Sn-incorporation effect on the covalent characteristics of Ga-O and Sn-O and the electronic structure near valence and conduction-band edges.
The introduction and experimental sections of the manuscript are satisfactorily prepared. However, the result and discussion parts have materials to be improved/added before publication in Nanomaterials, which are indicated as follows.
(1) In subsection 2.2, information on the crystal structures of the prepared films are required.
(2) In subsection 3.3, a discussion on correlation between the surface morphology given in subsection 3.2 and the optical parameters (refractive index and extinction coefficient) are recommended.
(3) In subsection 3.4, the definition of the energy-loss peak function is helpful for reader understanding.
(4) In subsection 3.5, the figure for the MOS structure used for I-V and C-V measurements are recommended.
(5) In subsection 3.5, if the authors have determined the carrier type of the films prepared in this study, the evidences and argumentations on the type and concentrations are required.
(6) As mentioned in introduction section, Refs. 15, 16, and 17 used Si, Ge, and Sn are as doping elements, respectively, for Ga2O3, Comparison between these results and the present results are strongly recommended.
